# The Adaptation of Internet of Things in the Indian Insurance Industry—Reviewing the Challenges and Potential Solutions

Maryam Saeed [1], Noman Arshed [2] and Haikuan Zhang [3,4,*]

1   Department of Banking and Finance, Dr Hasan Murad School of Management, University of Management & Technology, Lahore 54700, Pakistan; f201713002@umt.edu.pk
2   Economics, Department of Economics and Statistics, Dr Hasan Murad School of Management, University of Management and Technology, Lahore 54700, Pakistan; noman.arshed@umt.edu.pk
3   School of Innovation and Entrepreneurship, Entrepreneurship Institute, Guangzhou University, Guangzhou 510006, China
4   School of Economics and Management, East China University of Technology, Nanchang 330013, China
*   Correspondence: zhk1983@126.com

**Abstract:** The concept of insurance was found several centuries before Christ. Correspondingly, Chinese and Babylonian traders practiced moving or dispensing risks in the second and third millennia BC. Nowadays, insurance is the backbone of the economy. The recent introduction of big data, IoT, and other forms of InsurTech led to the fourth industrial revolution in insurance in the developed world. The industry is looking to improve the ergonomics of remote sensing technology to improve the acceptability of the clients. The adaptation of IoT in developing economies may provide a solution in increasing insurance penetration. This study explores the challenges and solutions in adopting IoT to increase insurance penetration in India. This study applied a systematic literature review (SLR) to extract the themes/variables related to challenges and solutions in adopting IoT in India's insurance sector. Several keywords were used to search the relevant literature from Google Scholar. Based on inclusion and exclusion criteria, the filtered studies were explored. This study listed several challenges and their solutions in the adaption of IoT in the Indian insurance industry. Policymakers could adapt the suggestions provided to improve the service delivery insurance sector. The authors listed several challenges and solutions in the adaption of IoT in the Indian insurance industry through a systematic literature review to facilitate the policymakers to make the right decisions.

**Keywords:** InsurTech; IoT; cloud computing; insurance industry; digital technologies





## 1. Introduction

IoT makes a connection system among billions of people and physical objects globally by the internet. This concept has received much attention due to the high internet penetration rate consequent to the goal of the Indian government to facilitate the Indian digital reserves for IoT reaching USD 15 billion by 2021 and USD 560 billion by 2022 at a global level according to a report by Parnesh [1]. Indeed, 5G technology will boost in subscription of IoT devices. According to insurance experts, 5G technology will boost the application of vehicle telematics, causing growing demand and supply of auto insurance. COVID-19 also greatly impacts growing demands for IoT solutions, particularly health administration. IoT enabled insurance services during the coronavirus pandemic to increase usage-based insurance (UBI). To improve the IoT subscription, insurers need to adopt machine learning, telematics, and social media (e.g., Reportlinker [2]). Mikhail [3] revealed that wearable technology such as FitBit is used to help people track their health details constantly, which can be used by doctors for diagnosing and treating patients, and health insurance enterprises give a rebate on premium policyholders on the use of this kind of technology [4]. Sensors or detectors linked with the internet helps in detecting smoke in case of fire in a building, and

telematics helps monitor automobiles' speed and the behavior of a driver, which is useful for insurance claim disbursement. There are some challenges in adopting IoT technology in India, which will be explored in this study, along with solutions that will help the insurance companies increase the insurance penetration level in India. Accordingly, the problem statement is as follows:

What are the challenges and solutions in adopting the internet of things (IoT) in the Indian insurance industry to enhance insurance penetration?

The following are research questions to answer the above problem statement:

Q1. What are the challenges in adopting IoT (internet of thing) in academic studies conducted in the Indian context?

Q2. What are the solutions in adopting the internet of things (IoT) in academic studies conducted in the Indian context?

Several studies in the Indian context have discussed the challenges in the adaption of the IoT in the Indian insurance sector, including studies by Bishwajit et al. [5] and Raviteja and Mansi [6]. These studies have listed several problems and challenges in transforming the insurance sector.

These listed studies have explored the problems in the specific domain. However, there is a lack of studies in connecting the problems with potential solutions. This study used the SLR method to integrate the problems mentioned and the solutions proposed.

This paper is divided into several sections—namely, Introduction, Materials and Methods for analyzing and describing reviewed findings, Policy Implication, Limitations, Future Direction, and Discussion and Conclusions.

## 2. Materials and Methods

In the current study, the SLR approach was selected to answer research objectives based on the above evidence. Due to the qualitative and exploratory nature of these studies, quantitative econometric techniques could not be applied to derive challenges and solutions in the Indian context so that the Indian insurance industry can adopt solutions if facing similar challenges. Figure 1 shows the IoT-based smart farming, in which figures denote its entire implementation procedure.

Many researchers have applied SLR and published papers in international journals. SLR consists of steps listed in Figure 2, which were applied to extract and read relevant papers systematically. As evidence, the following papers applied the SLR method using publications in recognized journals:

Review of Technology Adaption frameworks in Mobile Commerce: This study reviewed 201 articles and adopted a systematic literature review to analyze and highlight the usage of technology adaption theories in mobile commerce.

Barriers to Information Technology Adaption within Small and Medium Enterprises: A Systematic Literature Review: This paper aimed to create a systematic literature review to understand barriers to IT adaption within SMEs better. Based on 132 selected studies, this study identified 18 barriers, categorized according to internal and external parameters.

All primary studies in the literature such as conference proceedings reported challenges and solutions in adopting IoT, which are listed in Table 1.

Table 1 explains the criteria opted for including or excluding the available literature. In this criteria, this study opted for full articles and conference proceedings on relevant topics from 2015 to 2021 and excluded reports or thesis or website content.

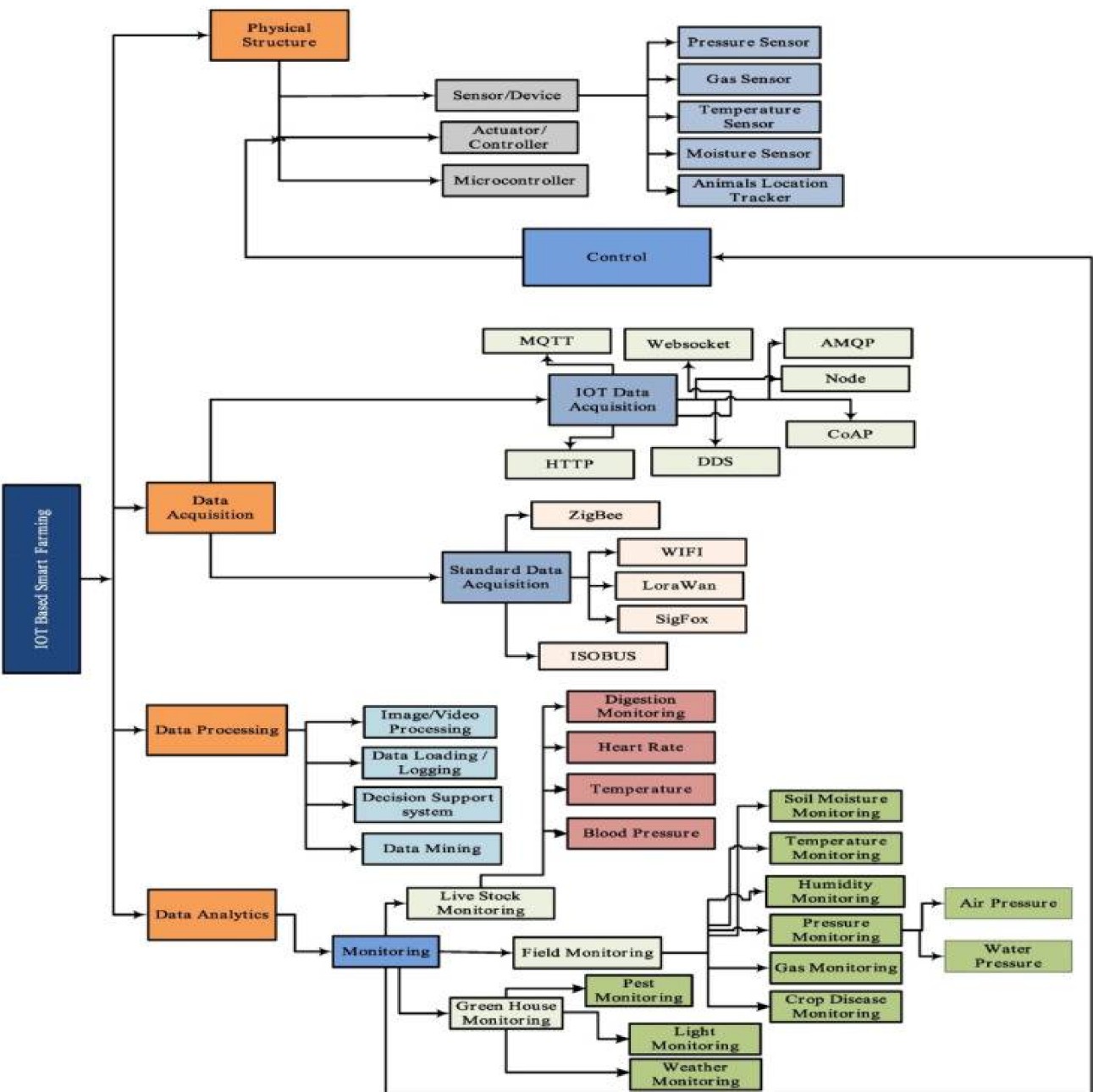

**Figure 1.** IoT smart farming framework Farooq et al. [7].

### 2.1. Search Strategy and Selection Process

Research paper publication platforms such as Google Scholar and Emerald were used for this review as academic search engines. Then, a combination of the following search terms was applied: IoT* AND (health insurance* OR insurance sector*) AND (challenge* OR obstacle* OR issue* OR disadvantage* OR threat). The search was conducted between 2015 and 2020. The filtering process of studies retrieved from the Google Scholar and Emerald databases was performed in three phases, as shown in Figure 2.

### 2.2. Data Extraction and Data Synthesis

The two reviewers individually extracted the following data from the included articles: author name, year of publication, country of publication, publication type, and findings. Subsequently, a narrative synthesis of the extracted data was accomplished.

Tables 2 and 3 summarize the characteristics of reviewed journals addressing the Indian health insurance sector and the relevant challenges and solutions. From this summarized content, common repeated challenges and solutions were derived, as shown in Tables 4 and 5.

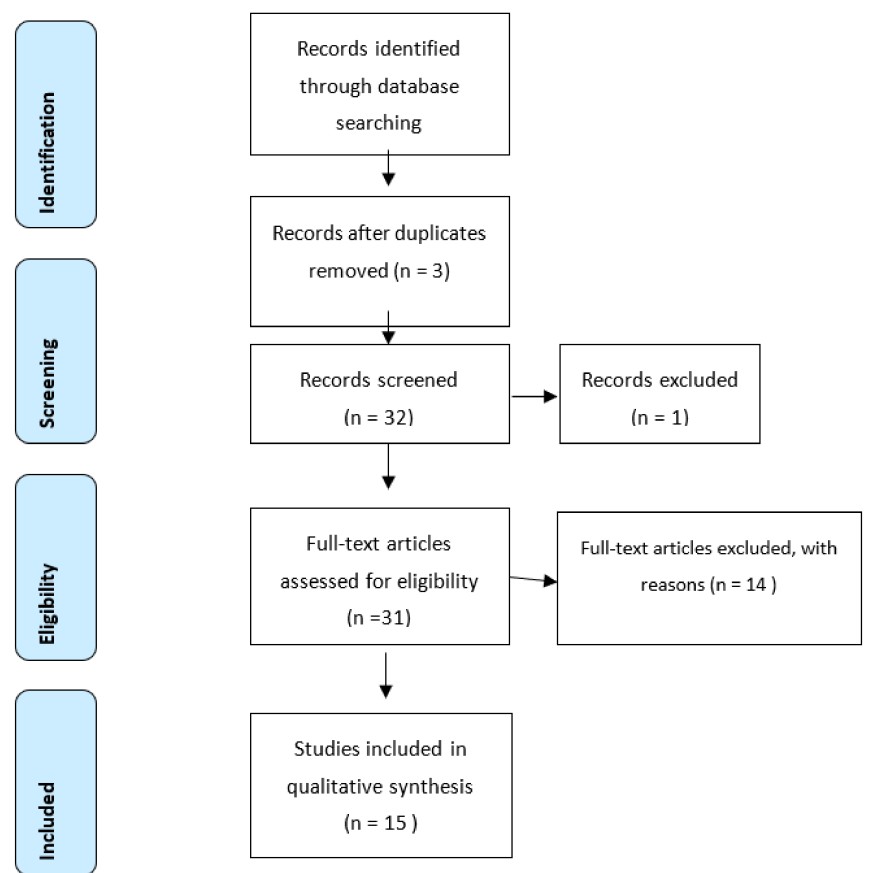

**Figure 2.** PRISMA (authors).

**Table 1.** Inclusion/exclusion criteria.

| Criteria | Specified Criteria |
|---|---|
| Inclusion | Literature and conference proceedings that address challenges and solutions in the adaption of IoT and InsurTech in the Indian financial sector, including the insurance sector and health insurance |
| | Studies available from 2015 onwards |
| | Primary and secondary studies |
| Exclusion | Studies stated in a language other than English |
| | Data from magazines, newspapers, thesis, reports |
| | Studies conducted in sectors other than the financial sector such as education, manufacturing |
| | Studies that merged big data analytics and other technologies |

**Table 2.** Characteristic of reviewed journals of Indian health insurance sector.

| No. | Journal Name/Conference Name | Paper Topic/Conference Paper Name | Methodology | Year | Author |
|---|---|---|---|---|---|
| 1 | International Journal of Engineering Research and Technology | Big Data, CEP and IoT: Redefining Holistic Healthcare Information Systems and Analytics | Exploratory | 2015 | Tawseef et al. [8] |
| 2 | Digital Policy, Regulation, and Governance | Regulation and Governance of the Internet of Things in India | | 2018 | Sheshadri and Arpan [9] |
| 3 | Humanities and Business Management conference | Challenges and Opportunities in Social Sciences | | 2019 | Saumya [10] |
| 4 | Second International Conference on Inventive Communication and Computational Technologies (ICICCT 2018) | A Secured IoT Based Webcare Healthcare Controlling System Using BSN | | 2018 | Kale and Bhagwat [11] |
| 5 | Multimedia Tools and Applications Journals | A Healthcare Monitoring System Using Random Forest and Internet of Things (IoT) | | 2019 | Pavleen et al. [12] |
| 6 | International Journal of Engineering Research and Technology | Big Data, CEP and IoT: Redefining Holistic Healthcare Information Systems and Analytics | | 2015 | Touseef et al. [8] |
| 7 | TIMSCDR Research Journal | Study of Wireless Sensor Network | | 2015 | Raviteja and Mansi [6] |
| 8 | Diabetes and Metabolic Syndrome: Clinical Research and Reviews | Internet of Thing (IoT) Applications to Fight Against COVID-19 Pandemic | SLR | 2020 | Ravi et al. [13] |
| 9 | Business Strategy and Development | Democratizing Health Insurance Services, Accelerating Social Inclusion through Technology Policy of Health Insurance Firms | | 2019 | Bishwati [5] |
| 10 | International Journal of Scientific Research in Computer Science Applications and Management Studies | Human Activity Recognition Using IOT Challenges and Opportunities | | 2018 | Vipin and Suchithra [14] |
| 11 | Challenges of IoT in Healthcare | IoT and ICT for Healthcare Applications | | 2020 | Nishu and Sara [15] |
| 12 | International Journal of Pervasive Computing and Communications | IoT Role in the Prevention of COVID-19 and Healthcare Workforces Behavioral Intention in India-An Empirical Examination | Quantitative | 2020 | Vijay et al. [16] |
| 13 | Enterprise Information Systems | IoTPulse: Machine Learning-Based Enterprise Health Information System to Predict Alcohol Addiction in Punjab (India) Using IoT and Fog Computing | Exploratory | 2020 | Arwinder [17] |
| 14 | International Conference on Computer Communication and Informatics | Survey On Security of IoT | | 2020 | Vinitha and Mohanapriya [18] |
| 15 | Management Journal of Siva Sivani Institute of Management | Fintech Services in India: Issues and Challenges | | 2018 | Ramana [19] |

Table 3. IoT adaption in Indian insurance industry: issues and solutions.

| No. of Studies | Issues | Solutions |
|---|---|---|
| Study 1 | • This technology is in the infancy stage in India | • Blockchain technology assists healthcare providers in the security of data exchange |
| Study 2 | • Lack of research and development in this IoT technology<br>• Lack of good governance and regulation<br>• Lack of awareness<br>• Security issue<br>• Privacy issue<br>• Lack of fund | • The policy is vital for the promotion of IoT in India<br>• Government in making policies<br>• Short training programs can help in human resource development |
| Study 3 | • Processing speed issue for analyzing big data volume generated by IoT timely manner to make the right decision<br>• Data ownership issue as a customer may claim over the right of their personal data<br>• Historical data of claim history to switch insurer at renewal time<br>• Lack of regulation<br>• Cyberattacks | • Data management strategy |
| Study 4 | • Data privacy issue regarding leaking patient illness<br>• Data integrity issue due to the absence of a trustworthiness system | Nil |
| Study 5 | • Security of data | Nil |
| Study 6 | • Implementation of this technology is infancy stage of this technology in India | Nil |
| Study 7 | • Topology management complexity and node distribution<br>• Node cost<br>• Immense scalability is required due to networking<br>• Sensor security issues<br>• Remote management of a sensor network makes it virtually impossible to detect physical tampering<br>• Physical maintenance issue | • Blockchain technology can solve privacy and security, and traceability relating issues |
| Study 8 | • Security issues regarding patient data, which can be misused<br>• Cybercrimes issue<br>• Network integration issue due to involvement of different devices having different network<br>• Protocols that create difficulty in processing of data aggregation | • Cloud storage technology can enable to overcome the data storage problem |

**Table 3.** *Cont.*

| No. of Studies | Issues | Solutions |
|---|---|---|
| Study 9 | • Security issue<br>• Leaking privacy of individual data<br>• Technology procurement issue<br>• Lack of highly skilled personnel<br>• Lack of integrated technology platform<br>• Lack of regulation | • Designing organizational technology strategy<br>• A firm can enable value creation via technology<br>• Procurement of high-quality technology from vendor or partner supplying technology<br>• Recruiting tech-savvy employees<br>• Customer focus through technology |
| Study 10 | • Security issue<br>• Data alteration by a hacker<br>• Interoperability of IoT systems<br>• Sluggish processing time<br>• Need of high-performing computing system<br>• Lack of expertise related to cyber security<br>• Lack of stakeholder collaboration | • Usage of encryption to prevent unauthorized access of patient data<br>• IoT web portal should be protected with a strong password and authentication protocol |
| Study 11 | • Acquirement of data<br>• Latency issue while handling the giant volume of information | Nil |
| Study 12 | • Security and privacy<br>• Interoperability | Nil |
| Study 13 | • Network bandwidth<br>• Latency issue due to increasing number of patient<br>• Energy consumption | Nil |
| Study 14 | • Sensing layer security problems such as hub capturing, false data injection attack, booting attacks, malevolent code injection attack, spying and interference, and lack-of-sleep attacks; network layer security issues such as site attacks, steering attacks, information transit attacks, DDoS/DoS attack, access attacks; middleware layer security issues such as Mark wrapping attacks, SQL injection attack, man-in-the-middle attacks, and flooding attack in the cloud; gateway security issue such as end-to-end encryption, additional interfaces, secure onboarding, firmware refreshes; application-layer security issues such as information attacks, access control attacks, sniffing attacks, administration interruption attacks, reconstruct attacks | Nil |
| Study 15 | • Insufficient telecom infrastructure<br>• Lack of literacy<br>• Lack of fund<br>• Gender disparity issues such as restricted internet, cell phones access among women, especially in small towns and villages | Nil |

**Table 4.** IoT adaption challenges in Indian insurance industry.

| Challenges for IoT Adaption in India |
| --- |
| • Infancy stage of IoT implementation<br>• Lack of research and development in this IoT technology<br>• Lack of good governance in terms of standards, and law and regulation regarding data sharing via IoT devices<br>• Lack of awareness about IoT design and application<br>• Data security issue due to physical tampering or leakage while communication/exchanges data in IoT devices<br>• Limited processing speed or memory for analyzing big data volume generated by IoT<br>• Data ownership issue<br>• Data integrity issue<br>• Topology management complexity/network integration issue due to different devices having different networks<br>• Limited scalability issue<br>• Physical maintenance issue<br>• Protocols that create difficulty in processing of data aggregation<br>• Technology procurement issue<br>• Lack of highly skilled personnel<br>• Energy consumption |

**Table 5.** IoT adaption solutions in Indian insurance industry.

| Proposed Solutions for IoT Adaption in India |
| --- |
| • Blockchain technology assist in the security and privacy of data exchange<br>• Comprehensive, effective, implementable, and simple policy for IoT is essential for the promotion of IoT in India<br>• Collaboration is needed among government, industries, and academia, and a high-level advisory committee to arrange, develop, design, and test IoT devices in different sectors including health insurance<br>• Formulation of attack-resistant solutions to protect IoT devices from attack<br>• Data management strategy should provide unified solutions, tools, methodologies, and workflows for managing IoT data as core assets<br>• Recruiting tech-savvy employees or planning a short training program in human resource development<br>• Cloud storage technology can enable to overcome the data storage problem<br>• Procurement of high-quality technology from vendor or partner supplying technology<br>• Usage of encryption while data exchange in IoT system to prevent unauthorized access of patient data<br>• Removal of surcharges on electronic transactions<br>• Tax benefits for consumers and businesses using e-payment<br>• Establishing an organization technology structure for promoting partners and strategic allies for sharing information of interfaces<br>• Lack of trust |

*2.3. Findings of the Included Studies*

The following are the findings extracted from the studies:

The first challenge is that the implementation of IoT technology in India is in its infancy. The second challenge is the lack of research and development in IoT technology, as indicated by Bishwajit et al. [5], in areas such as hardware to software, robotics, artificial intelligence, internet security, digital payment systems, data storage, data encryption technology, and data transmission technology [20].

The third challenge is the lack of good governance and laws and regulations. IoT devices or software development are designed by ignoring security laws. Fewer efforts are made in designing standards and protocols regarding information sharing. Nallapaneni

and Pradeep [21] found the absence of regulations on the usage of IoT information and liability issues. Olakunle and Igbafe [22] found that regulations regarding controlling and ownership of farm information between farmers and data companies such as insurance companies should be settled down in agriculture. Standard legal systems are needed for upholding IoT device compatibility. There is a lack of standard configurations for interfacing many IoT devices, as well as an absence of internationally recognized laws and standards regarding data collection, sharing, and usage of human activity, according to the findings of Vipin and Suchithra [14].

The fourth challenge is a lack of awareness. Insurance companies have less awareness of the technical aspects of design risk protocols and policy plans with nonstop progression in IoT. The fifth challenge is security issues due to the chance of leakage of insurance data by IoT devices gathering valuable information, with diverse, prevailing technologies, and transferring that information to other IoT devices. Information security laws in India are still emerging. This intensifies the need to develop a technology policy, as indicated by Pavleen et al. [12]. Additionally, sensor security issues due to unreliable communication depending on packet routing were highlighted in the findings of Raviteja and Mansi [6]. In health insurance, security issues regarding patient data can be misused, as revealed by the findings of Ravi et al. [13]. Bishwati et al. [5] found that the security of big data produced is vital to extract significant insights. IoT security issues due to difficulty in identifying many authorized devices and managing them are indicated in the findings of Vipin and Suchithra [14]. Security issues exist regarding patient-sensitive data flowing in a network without encryption, which can be hacked for blackmailing, as revealed by Nishu and Sara [15]. Sensing layer security problems such as hub capturing, false data injection attacks, and booting attacks, as well as network layer security issues such as website attacks are highlighted in the findings of Vinitha and Mohanapriya [18]. Data security issues due to IoT-based systems depending on internet connectivity led to cyberattacks, according to the findings of Saumya [10]. Security issues in agriculture led to data loss and physical tamperings such as theft or attacks by animals or predators. Remote management of a sensor network makes it difficult to identify physical tampering. Olakunle and Igbafe [22] found that gateways are prone to congestion attacks, denial of service (DoS), and forwarding attacks.

The sixth challenge is processing speed, i.e., analyzing a volume of big data generated by IoT in a timely manner to make the right decisions. Network integration issues, due to the involvement of different devices having different networks and protocols, create difficulty in processing data aggregation, as indicated by Ravi et al. [13]. The limited processing power of IoT devices is highlighted in Vipin and Suchithra [14]. Data collected from different IoT devices increased processing time and the need for high-performance computing systems to process a massive amount of data, according to Nishu and Sara [15]. The seventh issue is data ownership, as customers may lay claim over the right of their personal data to switch insurers at the time of renewal, as per the findings of Saumya [10].

The eighth issue is data integrity due to the absence of a trustworthy system, as found by Kale and Bhagwat [11]. The ninth issue is topology management complexity due to node distribution, as indicated by Raviteja and Mansi [6]. The tenth issue is limited scalability, which is immense, due to networking among tens to hundreds or thousands of nodes and its application features and analytical capabilities, according to the findings of Raviteja and Mansi [6].

The eleventh issue is physical maintenance. It is necessary for better availability, utilization, and performance. The twelfth issue is existing gateways and protocols that need to support many IoT devices, as highlighted in the study of Olakunle and Igbafe [22]. The thirteenth issue is the technology procurement issue. The greatest challenges in IT procurement are related to supplier management. Ensuring the organization chooses the most reliable and appropriate supplier to deliver complex technology solutions is key to success, as is the procurement of high-quality technology from vendors or partners supplying the technology. The fourteenth issue is the lack of highly skilled personnel.

Lack of talent and training presents challenges for almost half of IoT adopters. Lack of expertise exists related to cyber security. The fifteenth issue is energy consumption. IoT gathers much information that needs to be processed and demands high energy to provide quality service.

For cyber security, attack-resistant solutions must be formulated appropriately to protect IoT devices from attacks [23,24]. Comprehensive, effective, implementable, and simple IoT policy is essential for promoting IoT in India. Collaboration is needed among government, industries, and academia, and a high-level advisory committee should be established to arrange, develop, design, and test IoT devices in the different sectors, including health insurance. There is also a need to establish an organizational structure that promotes creating partners and strategic allies such as customers, partners, and suppliers, and due to the dependency among ecosystems, stakeholders should establish a mechanism for sharing information and disseminating knowledge about interfaces, as indicated in the findings of Sudatta et al. [25]. Short training programs can help in human resource development. Trust can be developed in the adaption of IoT by training and awareness programs. IoT syllabus should be included in high school or undergraduate, graduate syllabus, creating linkages to industry, and funding for start-ups in the IoT area, as per Satya's findings [20]. Cloud storage technology can enable overcoming the data storage problem. In vehicular scenarios, cloud computing is not a good choice for task offloading since it is accompanied by large delays and low efficiency. Mobile edge computing (MEC) becomes a promising solution, due to its proximity to mobile vehicular terminals, and effectively reduces the task's transmission delay [26]. In remote sensing and Earth observation applications, ground objects represented by each hyperspectral image (HIS) pixel are composed of physical and chemical non-Euclidean structures, and HSI classification (HIC) is becoming a more challenging task. A convolutional network (DAGCN) can solve this challenge, which has a feature extraction method to extract deep abstract features and explore the internal relationship between HSI data [27]. The procurement of high-quality technology from vendors or partners supplying the technology, removing surcharges on electronic transactions, and providing tax benefits for consumers and businesses using e-payment will also promote IoT adaption.

### 2.4. Limitations

This study only reviewed past studies conducted in India, using the SLR technique. Most papers address the financial sector overall, but in the insurance sector, very rare studies are found from the Indian perspective. In addition, this study is qualitative, as it was only a review-based study in the Indian financial sector, including insurance.

### 2.5. Future Directions

This study can be extended by collecting primary data regarding IoT adaption challenges from experts in the Indian insurance sector. Future studies can collect primary and secondary data to compare other developing countries with similar human development indexes and cultures. This comparison will help India boost insurance penetration, which is very low, for example, 4.2% in FY21, with life insurance penetration at 3.2% and non-life insurance penetration at 1.0%. This is less than Western developed countries, with high insurance penetration and technology adoption rates.

### 2.6. Policy Implications

This study provided an overview of challenges and solutions to the Indian financial sector experts, including the insurance sector, who can understand and apply appropriate solutions if encountering similar challenges. These solutions will help adopt IoT and develop new insurance products such as usage-based vehicle insurance, which was not possible earlier; they will also help reduce false claims, giving an accurate picture of incidents. Further it can help in determination of correct insurance premium. This study

contributes to promoting insurance penetration in India, which lags in technology adoption and insurance penetration level, compared with developed western countries.

## 3. Discussion and Conclusions

This study explored the challenges and solutions in adopting IoT in the Indian insurance industry by using a systematic literature review. There were many challenges pointed out in the study. The synthesis linked several solutions with mentioned problems. A total of 26 studies were shortlisted in the domain of the Indian insurance sector, which were selected using appropriate keywords. The SLR method was applied to generate themes/variables, constraints, and solutions in adopting big data in the insurance sector. The derived challenges and solutions in the context of India's financial sector, including the insurance sector, were explained in detail, which practitioners can adopt to enhance insurance penetration, which is very low, compared with developed countries such as the US, with around 6% insurance penetration. Insurance penetration is directly linked with financial inclusion, which helps in poverty alleviation. This study is limited to the academic literature, which can be extended by taking primary interviews from Indian insurance experts and FinTech/InsurTech experts working at executive-level positions.

This study is helpful for the Indian insurance sector, which can understand the different challenges and adopt solutions recommended in this study to enhance insurance innovation, insurance penetration, and customer satisfaction, which are at much lower levels, compared with developed countries and other developing countries. This study is also helpful for countries that match the Indian cultural system and its human development index.

**Author Contributions:** Data curation, formal analysis, writing—original draft preparation, M.S.; conceptualization, supervision, writing—review and editing, N.A.; validation, investigation, writing—review and editing, H.Z. All authors have read and agreed to the published version of the manuscript.

**Funding:** This research received no external funding.

**Data Availability Statement:** This study has conducted SLR on available literature. So the data is the articles used which are cited in references.

**Conflicts of Interest:** The authors declare no conflict of interest.

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
