# Peer review of "The Adaptation of Internet of Things in the Indian Insurance Industry—Reviewing the Challenges and Potential Solutions"

_electronics, doi:10.3390/electronics11030419_

Round 1
Reviewer 1 Report
Comment1:
Some grammar and spelling errors exist in the paper, Please correct all the grammatical and syntax errors.
Comment2:
The paper needs more proofreading. For example, the end of the introduction part should describe the overall organization of the paper in the following for more clear structure.
Comment3:
professional terms should be explained when they first appear.
Comment4:
The figure in this paper should be clearer and explained.
Comment5:
More discussions on the merits and limitations of the proposed method can be further presented. For this purpose, the listed two papers may help.
DOI: 10.1109/JSTARS.2021.3079103
DOI: 10.1109/ACCESS.2020.2968339
Author Response
Thank you very much for your valuable comments. Authors have incorporated them with best of their ability

Reviewer 2 Report
The related work on deep area, like the method used in this area should be given. Some of tables or simulaiton results or comparing analysis should be added. CUrrently, only sentences are used to summary the topics.
The future works should be given and pointed well.
The edge computing, AI, applied in this area should be given as well, like the following related papers:
Jing Bai,Bixiu Ding, Zhu Xiao, Licheng Jiao, Hongyang Chen, Amelia C.
Regan, Hyperspectral Image Classification Based on Deep Attention Graph
Convolutional Network, to appear, IEEE Transactions on Geoscience and
Remote Sensing
Zhu Xiao, Xingxia Dai, Hongbo Jiang, Dong Wang, Hongyang Chen, Liang
Yang, and Fanzi Zeng, “Vehicular Task Offloading via Heat-Aware MEC
Cooperation Using Game-Theoretic Method,” IEEE Internet of Things
Journal, vol. 7, no. 3, pp. 2038-2052, Mar. 2020.
Author Response

(The authors gave the same response as above.)

Reviewer 3 Report
The paper is poorly written and requires much editing, the novelty of the paper is low.
Authors just created a table of references and basic solutions. I don't understand what readers will learn after reading this paper ?
It look like a class note. It can be treated as review paper for some workshop.
Author Response

(The authors gave the same response as above.)

Reviewer 4 Report
The authors listed several challenges and solutions in the adoption of IoT in Indian insurance industry through a systematic literature review, in order to facilitate the policymakers to make right decisions. However, there are some shortcomings need to be corrected.
- Grammatical mistakes
- In page 8, Table 3, Study 7, the phrase “packet based” should be an adjective in the sentence “sensor security issues...”, so it is better to add a hyphen between two words. The same errors appear later in the article, such as “high performance” in Table 3, Study 10 and paragraph 3, 4 in page 13.
- In page 10, Table 3, Study 13, the word “patient” should be “patients”.
- Formatting
- It is suggested to adjust the clarity of Figure 1. The text in figure 1 is too fuzzy to read after conversion to PDF format.
- Table 1 overlaps the number of rows and it’s hard to read.
- Description
- It is been noticed that a description “Figure 7” in section 2.1, page 4, however there is no corresponding figure in this article.
- There is lack of the corresponding text description for the tables that appears in this article and it’s better to add explanation to enhance readability.
- Improve the overall layout.
In addition, there is a confusion about the words “adaption” and “adoption”, because the authors use “adaption” in title, however they use “adoption” later in the abstract and header of Table 3 under same semantic.
Author Response

(The authors gave the same response as above.)

Round 2
Reviewer 2 Report
the english writing could be further improved. after it, it could be accepted for publication.
Author Response
Thank you very much for valuable comments. Authors have tried with best of their ability to incorporate these suggestions

Reviewer 3 Report
Authors updated the paper and make it suitable for the journal. Related work can be extend with below articles:
https://ieeexplore.ieee.org/abstract/document/9250428
https://ieeexplore.ieee.org/document/8913631
https://link.springer.com/article/10.1007/s12652-021-03459-4
Author Response

(The authors gave the same response as above.)
